# Gas adsorption and framework flexibility of CALF-20 explored via experiments and simulations

Rama Oktavian [1,5], Ruben Goeminne [2,5], Lawson T. Glasby[1], Ping Song[3], Racheal Huynh[3], Omid Taheri Qazvini[3], Omid Ghaffari-Nik[3], Nima Masoumifard[3], Joan L. Cordiner [1], Pierre Hovington[3], Veronique Van Speybroeck [2] & Peyman Z. Moghadam [4] ✉

In 2021, Svante, in collaboration with BASF, reported successful scale up of CALF-20 production, a stable MOF with high capacity for post-combustion $CO_2$ capture which exhibits remarkable stability towards water. CALF-20's success story in the MOF commercialisation space provides new thinking about appropriate structural and adsorptive metrics important for $CO_2$ capture. Here, we combine atomistic-level simulations with experiments to study adsorptive properties of CALF-20 and shed light on its flexible crystal structure. We compare measured and predicted $CO_2$ and water adsorption isotherms and explain the role of water-framework interactions and hydrogen bonding networks in CALF-20's hydrophobic behaviour. Furthermore, regular and enhanced sampling molecular dynamics simulations are performed with both density-functional theory (DFT) and machine learning potentials (MLPs) trained to DFT energies and forces. From these simulations, the effects of adsorption-induced flexibility in CALF-20 are uncovered. We envisage this work would encourage development of other MOF materials useful for $CO_2$ capture applications in humid conditions.

Metal-organic frameworks (MOFs), one of the most exciting developments in recent porous materials science, are now, more than ever, the center of attention as they make their way successfully into industrial applications for gas adsorption and separation processes[1–5]. Most adsorption applications, especially $CO_2$ capture from either post combustion flue gas or direct air capture are inevitably operated under humid conditions where many MOFs suffer from competitive adsorption of water[6–11]. In 2014, Shimizu's laboratory at the University of Calgary reported a Zn-based MOF named Calgary Framework 20 (CALF-20) for physisorptive $CO_2$ capture under real flue gas conditions[12]. CALF-20 demonstrates an excellent $CO_2$ adsorption capacity of 2.6 mmol/g at 0.15 bar and 298 K, $CO_2$ selectivity against water of up to 40% relative humidity, as well as durability and stability towards steam, wet acid gases, and prolonged exposure to direct flue gas stream[13].

CALF-20, shown in Fig. 1a, consists of repeating layers of 1,2,4-triazoles connected by Zn atoms with oxalate ions bridging the layers. 1,2,4-triazole is well known for its water and basic environment stability; and its geometric rigidity, strong binding affinity, and high basicity has been exploited for constructing other robust MOFs[14–20]. Moreover, the scalability of CALF-20's synthesis has been demonstrated due to its relatively benign synthesis conditions. The use of methanol and water as solvents, as well as low-cost, commercially available starting materials results in high product yields of up to 90% and an exceptional space-time yield of 550 kg/m³ day[13].

[1]Department of Chemical and Biological Engineering, The University of Sheffield, Sheffield S1 3JD, UK. [2]Center for Molecular Modeling (CMM), Ghent University, Technologiepark 46, 9052 Zwijnaarde, Belgium. [3]Svante Inc., 8800 Glenlyon Pkwy, Burnaby, BC V5J 5K3, Canada. [4]Department of Chemical Engineering, University College London, London WC1E 7JE, UK. [5]These authors contributed equally: Rama Oktavian, Ruben Goeminne. ✉e-mail: p.moghadam@ucl.ac.uk

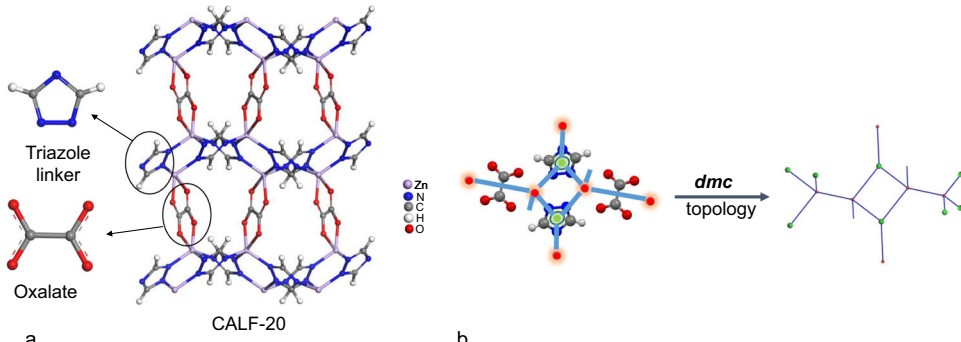

Fig. 1 | **The structure and topology of CALF-20. a** 3D representation of CALF-20 along with its azolate linker and oxalate pillar. **b** A schematic diagram showing the simplification of CALF-20 into its underlying dmc topology. The triazolate and oxalate linkers are disconnected from the metal nodes and simplified into 3-c and straight-through branches, respectively. The red spheres represent metallic nodes and the green spheres represent organic nodes, connected via blue linker 'branches'.

## Results and discussion

### Geometric characterization and gas adsorption properties of CALF-20

In 2021, Svante reported single-step commercial synthesis of CALF-20 for Temperature Swing Adsorption (TSA) processing of up to 1 tonne of $CO_2$ removal per day from cement flue gas[21]. CALF-20 has now become the hallmark of success among MOFs undergoing the scale up process from laboratory to industry - as it ticks many of the required boxes for an optimum adsorbent for practical $CO_2$ capture. Developing new materials for $CO_2$ abatement has never been more critical, and in this context, a number of research groups have started examining CALF-20 in more detail with the aim of aiding the design and development of other adsorbent materials for selective adsorption of $CO_2$. In late 2023, Ho and Paesani[22] and Magnin et al.[23] studied competitive adsorption and diffusion of water and $CO_2$ in CALF-20 via classical Monte Carlo and molecular dynamics simulations. At the same time, Chen et al.[24] looked into structural transformations of CALF-20 in humid environments via powder X-ray diffraction (PXRD) and pair distribution function analysis. In the present study, in collaboration with MOF scientists from Svante, we used a close feedback loop between simulations and experiments to obtain molecular-level insights into some of the key $CO_2$ adsorption properties of CALF-20, and determine its structural flexibility triggered by the presence of guest molecules. We investigated water-CALF-20 interactions through water adsorption simulations and hydrogen-bond analysis, and studied the structural transformations of CALF-20 using in-situ adsorption/PXRD data in combination with first-principles molecular dynamics (MD) simulations. With the aid of machine learning potentials (MLPs) trained to DFT data, fully converged free energy profiles of both the empty and guest-loaded frameworks were also computed, demonstrating the guest-induced flexibility of the framework. Importantly, in this work, we note that all the computations were performed before laboratory synthesis and physical gas adsorption measurements were carried out by Svante. The excellent agreement between simulation and experiment provided a powerful example of the predictive ability of molecular modeling, showcased in the assessment of MOF candidates for $CO_2$ capture in wet conditions.

## Results and discussion

### Geometric characterization and gas adsorption properties of CALF-20

To perform the simulations in this work, we used the Crystallographic Information File (CIF) from the published CALF-20 structure[13]. We first calculated the geometric properties of CALF-20 such as the surface area, pore size distribution, largest cavity diameter (LCD), pore limiting diameter (PLD) and topology. We note that computational characterization of geometric properties can provide valuable information about the expected adsorption performance of materials and help to draw deeper insights from the experimental observations. The LCD and PLD values in CALF-20 are ca. 4.3 Å and 3 Å, respectively: pore size ranges that provide a tight fit for $CO_2$ adsorption. Figure 1b shows the characterization of CALF-20's topology. The structure can be separated into $C_2N_3$ (triazolate) and $C_2O_4$ (oxalate) linkers, with individual zinc atoms as the metal nodes. After considering these two linker types, we simplified the structure using the SingleNode approach and arrived at the Reticular Chemistry Structure Resource (RCSR) dmc topology, calculated using ToposPro[25] and CrystalNets[26].

We also simulated the $N_2$ adsorption isotherm at 77 K and compared it with experimental measurements (Fig. 2a). We found excellent agreement between the two isotherms proving that the synthesized sample was highly crystalline and successfully activated. By strictly following the BET consistency criteria[27,28], we obtained the BET area of 550 $m^2$/g for CALF-20. Figure 2b shows simulated $CO_2$ adsorption isotherms for CALF-20 at a range of temperatures from 273 K to 387 K and compares them with experiments conducted at Svante. Overall, the predicted $CO_2$ uptakes are in good agreement with experimental data for the entire pressure range and across different temperatures. Flue gas typically consists of about 0.1–0.15 atmospheres of $CO_2$ pressure, and at these conditions, the amount of $CO_2$ adsorbed is ca. 0.4, 0.7, 1.3, 1.9, and 3.3 mmol/g at 387, 365, 343, 323, and 273 K respectively. One interesting observation is that the $CO_2$ adsorption predictions are slightly lower than measurements at pressures higher than ca. 150 mbar especially for the isotherm obtained at 273 K. Generally, the experimentally synthesized MOFs contain solvents in their pores, which can be removed upon activation. Before performing gas adsorption simulations, these solvent molecules can be fully removed mimicking the experimental activation process. This process assumes that the experimental activation is successful in removing all residual solvent inside the pores and the structure is not changed upon removing the solvent. Clearly, incomplete experimental activation in MOFs can reduce the accessibility to the pore space. Therefore, when solvent-free structures are used in simulations, the amount of predicted gas adsorption is usually higher than experimental measurements, given that more pore space is accessible for guest molecules[29,30]. Since the atoms in CALF-20's structure are fixed at their crystallographic positions during the simulations, underpredicting $CO_2$ uptake indicated to us that CALF-20 undergoes structural changes in the presence of external stimuli such as gas adsorption or temperature. Moreover, we present here the QR code (https://p3d.in/RIA6W) for the augmented reality (AR) of $CO_2$ adsorption snapshot in CALF-20 adapted from our previous work[31] to visualize more clearly the interaction between $CO_2$ molecules and framework.

To investigate potential interactions involved for guest-induced structural dynamics of CALF-20, we monitored structural

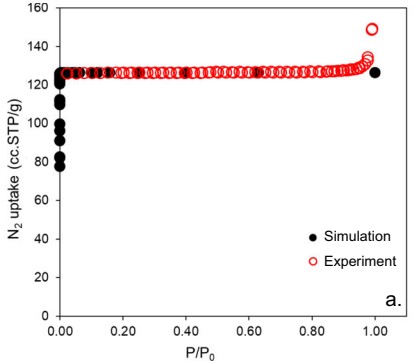

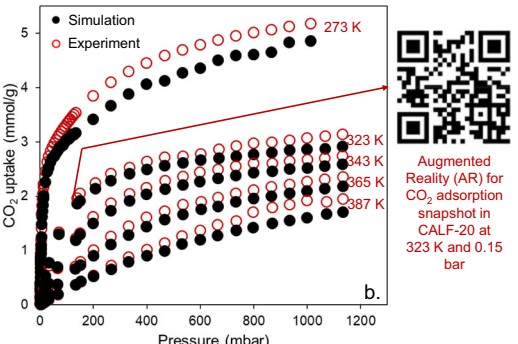

**Fig. 2 | N$_2$ and CO$_2$ adsorption isotherms in CALF-20. a** Experimental and simulated N$_2$ adsorption isotherms at 77 K in CALF-20. **b** Comparison between simulated and experimental CO$_2$ adsorption isotherms for CALF-20 at different temperatures along with QR code for augmented reality (AR) of CO$_2$ adsorption snapshot at 0.15 bar and 323 K. Black solid symbols represent simulation data, and red open symbols represent experiments carried out at Svante.

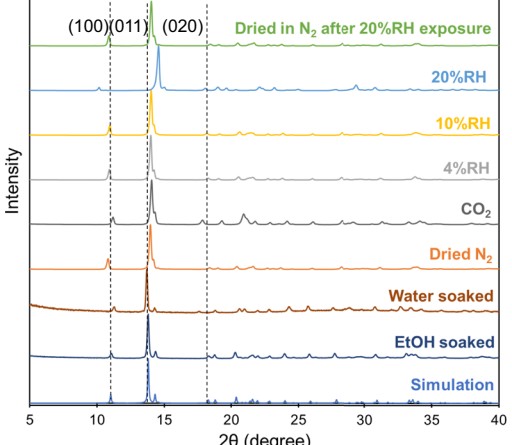

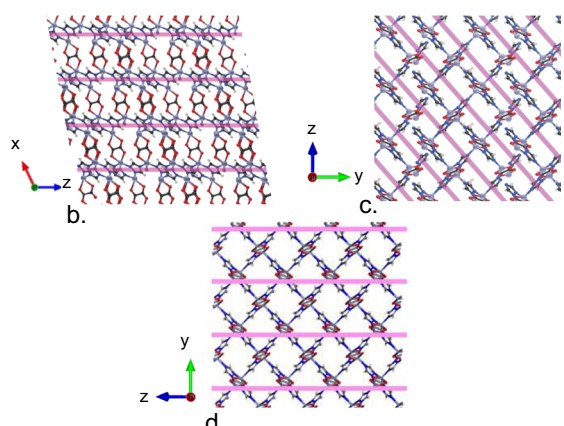

**Fig. 3 | CALF-20 structural changes characterized via in-situ adsorption/PXRD. a** Comparison of PXRD patterns for CALF-20 at different adsorption conditions, and for fully dried samples under dry N$_2$ and CO$_2$ at 50 °C. For the samples under gas flow, CALF-20 was first heated to 110 °C for 30 min under dry N$_2$ flow to evacuate the sample. Then, the sample was cooled to 50 °C and the PXRD pattern was collected under N$_2$ or CO$_2$ flow at 50 °C. PXRD patterns under different relative humidities (%RH) were all collected in N$_2$. Dashed lines represent the peaks for the simulated structure. CALF-20 structure with highlighted **b** (100), **c** (011) and **d** (020) *hkl* planes. The pink lines represent the relevant planes. Atoms coloring scheme is: red, oxygen; blue, nitrogen; white, hydrogen; gray, carbon, and light blue, zinc.

transformations via in-situ powder crystal X-ray diffraction (PXRD). We obtained PXRD data under different gas and liquid exposures (N$_2$, CO$_2$, water, and ethanol) and compared the results with the simulated CALF-20 (Fig. 3a).

The CIF from the published CALF-20 structure contains EtOH as guest molecules. Hence, when the experimentally synthesized CALF-20 powder is soaked in EtOH and then briefly dried (see Supplementary Methods 1 for details), the PXRD obtained matches very well with the simulated one. Since solvents induced substantial changes to the PXRD pattern, we evacuated CALF-20 by drying the powder in-situ at 110 °C for 45 min under N$_2$ flow (Fig. 3a), see Supplementary Methods 1 for more details.

The evacuated PXRD pattern is similar to the simulated or EtOH soaked sample, with only slight changes in the (100) and (011) reflections. The (100) plane in the simulated pattern, which corresponds to the interplanar distances between the Zn-triazolate layers in the xz-plane (Fig. 3b), exhibits a slight shift to lower 2θ and expansion of this plane when the sample is evacuated. There is a corresponding shift in the (011) reflection, which represents the middle of the pore (Fig. 3c), to a higher 2θ of the evacuated sample, suggesting a contraction along this plane. This implies that in the absence of solvent, the pores of CALF-20 as viewed along the x axis are contracted, with a corresponding expansion between the Zn-triazolate layers compared to the

EtOH soaked CALF-20. We take this evacuated CALF-20 as a baseline for further guest loaded studies.

Given CALF-20's ability to selectively adsorb CO$_2$ over water at low relative humidity (RH), we loaded CALF-20 with water and CO$_2$ to observe the effects on the PXRD pattern (see Supplementary Methods 1 for experimental details). Compared to the evacuated sample, loading CALF-20 with 100% CO$_2$ and soaking in liquid water induced a slight shift in the (100) reflection to higher 2θ, showing contraction between the Zn-triazolate layers, similar to what is observed in the EtOH soaked sample. In addition, the (020) reflection corresponding to the planes containing the oxalate moieties (Fig. 3d) shifts to lower 2θ for both water and CO$_2$, and it does not show much change for ethanol. In contrast, the (011) reflection remains the same in the CO$_2$ loaded sample, whereas a shift to lower 2θ is observed in the water soaked sample. The PXRD pattern of CALF-20 does not show significant changes at low RH (4% and 10%). At 20% RH, an obvious change of pattern was observed at (100), (011) and (020) planes. This is consistent with the reported phase change of CALF-20 in moisture[24]. Thus, H$_2$O accommodation requires the (011), (100) and (020) peaks to shift significantly, while CO$_2$ requires only the (100) and (020) reflections to move. The differences between PXRD patterns at 20%RH and water soaking indicated that water soaking might bring different changes to the structure compared to the adsorption of water vapor. Overall,

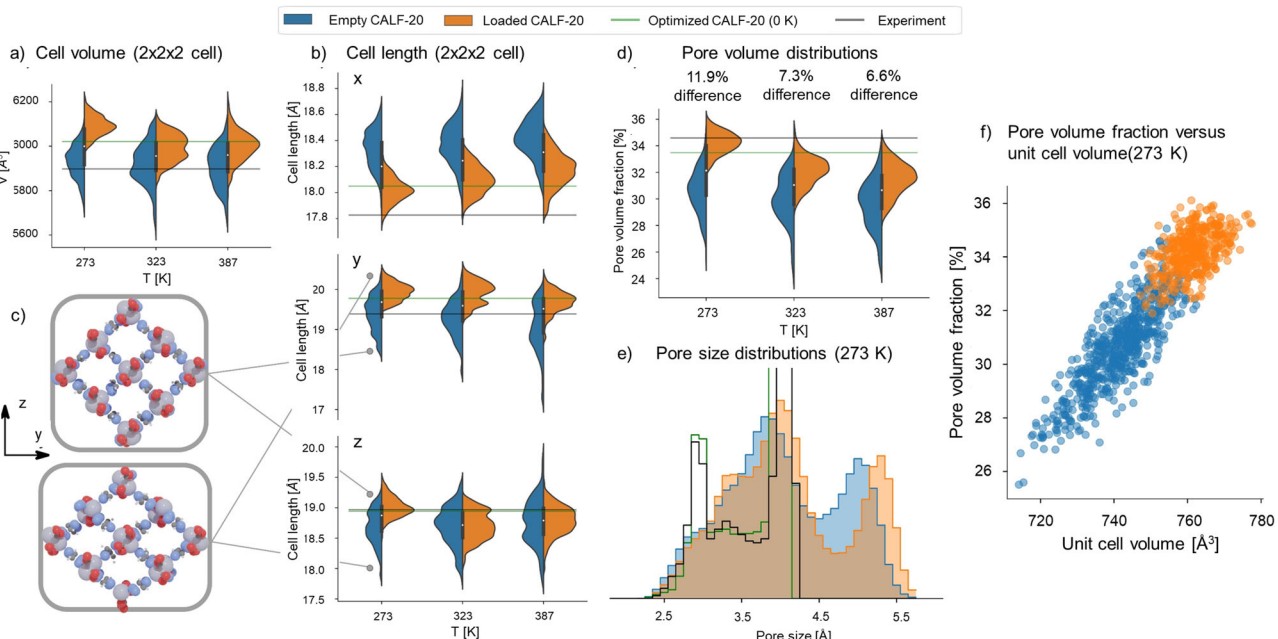

**Fig. 4 | CALF-20 structural changes characterized via first-principles molecular dynamics simulations.** Distributions of (**a**) cell volume (2x2x2) cell), **b** cell lengths, and **d** pore volume obtained from MD simulations of the empty (blue) and guest-loaded framework (orange) at 273 K, 323 K, and 387 K, compared to the values of the DFT optimized framework (green) and the experimentally resolved structure (black). **c** Two snapshots from MD simulations demonstrating the variability of the cell vectors perpendicular to the Zn-triazolate layers. Atom coloring scheme is: light gray, zinc; red, oxygen; white, hydrogen; blue: nitrogen; gray: carbon). **e** Pore size distributions of the experimental structure (black), the DFT optimized structure (green), and those averaged over MD trajectories of the empty and guest-loaded CALF-20 at 273 K. **f** Pore volume fraction as a function of the unit cell volume for the empty CALF-20 framework (blue) and the $CO_2$ loaded framework (orange) at 273 K.

water is able to bring more changes to the framework compared to all other guests discussed here. In other words, $CO_2$'s accommodation requires less structural change than $H_2O$; this could explain why CALF-20 accepts $CO_2$ over water at low RH. A simplified explanation would be that $CO_2$ adsorption into CALF-20 does not require much structural adjustment of the activated phase, whereas water needs the framework to open slightly more: even before there is sufficient water to make this structural change, the $CO_2$ fills up, making CALF-20 more $CO_2$ selective. These changes in the PXRD patterns demonstrate that CALF-20 is not rigid and undergoes structural flexibility dependent on the guests inside the pores.

Simulation of MOFs exhibiting structural flexibility is challenging. Our reported GCMC simulations of gas adsorption (Fig. 2) made the assumption that framework atoms are fixed at their crystallographic positions, thus we modeled CALF-20 as rigid. This assumption is valid for many MOFs whose building block topology do not allow for high degrees of flexibility. However, simulated gas adsorption predictions can deviate from experiments when MOFs are structurally flexible in response to external stimuli such as temperature or guest loading[32]. Here, to further investigate the effects of adsorption-induced flexibility in CALF-20, we also performed MD simulations at the density-functional (DFT) level of theory (PBE-D3(BJ)) using CP2K[33]. By comparing MD simulations of both the empty and the guest-loaded CALF-20 framework at experimentally observed loadings, the effect of guests on the framework can be directly determined. Simulations were performed in the NPT ensemble (controlling the temperature and pressure, allowing the cell shape to fluctuate) with a 2x2x2 supercell of CALF-20 at 273 K, 323 K, and 387 K for a duration of 20 ps with a time step of 1 fs. From these MD trajectories, we computed histograms of the cell volume, cell lengths, pore volumes and pore size distributions, as shown in Fig. 4.

Firstly, consider the cell lengths shown in Fig. 4b. For the empty framework there is most variability along the y and z-axis, perpendicular to the Zn-triazolate layers. To illustrate this, two snapshots of the framework are extracted from the MD simulations, the yz-plane of

which is shown in Fig. 4c, with cell lengths annotated on Fig. 4b. We can see there is significant flexibility where one of the cell lengths varies inversely with the other: this corroborates with the differences observed in the PXRD patterns of CALF-20 upon exposure to guests (Fig. 3). This flexible mode is significantly inhibited when guests are present in the framework, as can be seen from the narrower orange distributions in Fig. 4b.

Secondly, the previous conclusion can also be drawn from the volume histograms in Fig. 4a. The difference between the guest-loaded and empty framework is largest at 273 K, at which the most guests are adsorbed. The adsorbed guests hold open the framework, resulting in a volume of 6089 Å$^3$, compared to a volume of 5930 Å$^3$ for the empty CALF-20. Even though this difference of 2.7% appears quite small, the effect is much more pronounced when considering the pore volume fraction of the framework, shown in Fig. 4d. This fraction is equal to the volume accessible in the framework for a nitrogen probe divided by the total volume, as computed from PoreBlazer[34]. This fraction is a measure for the amount of space accessible for guest molecules. At 273 K, the difference between the pore volume fractions of the empty and loaded frameworks is 11.9%, compared to 7.3% at 323 K and 6.6% at 387 K. The adsorbed guests clearly increase the pore volume available in the framework, expanding the space for more adsorbates, consistent with the hypothesis that the difference between the experimental observations and the GCMC simulations at 273 K is due to the ability of the framework to adapt to the presence of guests.

Lastly, the pore size distributions at 273 K are shown in Fig. 4e. Comparing the empty and guest-loaded frameworks in blue and orange, it is observed that the presence of guests increases the size of the largest pores in the framework, holding the material open. The correlation between the cell volume and pore volume fraction of the empty and guest-loaded framework is shown in Fig. 4f. Large spreads across these quantities can be seen for both the empty and guest-loaded framework suggesting that a thorough characterization of the adsorption properties for both should include this variation.

## Empty and guest-loaded DFT free energy profiles

These initial results indicate that the framework shows a degree of flexibility upon guest adsorption. To determine the relative stability of the framework at different volumes in the presence of guest adsorbates and at different temperatures, the free energy profiles can be derived[35]. However, as the construction of these profiles at the PBE-D3(BJ) level of theory is computationally excessively demanding, machine learning potentials (MLPs) were used instead in the following way. First, short (2 ps) DFT metadynamics simulations were performed on the $2 \times 2 \times 2$ CALF-20 supercell with the cell volume as a collective variable to explore the space of possible states of the framework. Within these simulations, all relevant volumes of the framework were sampled. Then, short DFT umbrella sampling simulations lasting 0.5 ps were performed, restrained at volumes between 4000 and 7000 Å$^3$ with a step of 50 Å$^3$, and a temperature of 500 K. Snapshots are taken every 5 fs from these simulations and together make up the training set for the MLP (using NequIP[35]). The approach of generating enhanced sampling DFT MD data for training an MLP has been successfully applied before to model the flexibility of MOFs[36]. The training error on the MLP was 0.13 meV per atom on the energy and 28.3 meV/Å on the forces. With the trained MLP, longer (15 ps) umbrella sampling simulations were then performed at 223 K, 273 K, and 387 K again at volumes between 4000 and 7000 Å$^3$ with a step of 50 Å$^3$. The calculated free energy profiles at these three temperatures are shown in Fig. 5a.

The predicted unit cell volume at 273 K agrees very well with the experimentally obtained volume. Furthermore, temperature only has a moderate effect on the relative stability of the framework. In Fig. 5b, the internal pressure (negative derivative of the free energy) is shown as function of the unit cell volume. From the simulations, the existence of a metastable closed pore (cp) is predicted at a unit cell volume of 585 Å$^3$. This cp phase can be reached by applying a mechanical pressure larger than the transition pressure (376 MPa at 273 K). This metastable cp phase only disappears, with a transition back to the large pore (lp) being predicted, when lowering the mechanical pressure to below the cp−lp transition pressure (111 MPa at 273 K). Such phase transitions have not been observed experimentally, but this could be due to the large magnitude of the transition pressures required.

The method to predict the free energy profiles of the empty framework was also used to predict the free energy of the $CO_2$-loaded framework. Again, training data at a $CO_2$ loading of 15.7 cc/g, 31.5 cc/g, 63.0 cc/g, 94.5 cc/g, and 126 cc/g were generated, MLPs were trained, and MLP umbrella sampling simulations were performed. The resulting free energy profiles are shown in Fig. 5c. As expected from the results in Fig. 4, higher guest loadings make the space of accessible volumes narrower, as well as shifting the equilibrium unit cell volume upwards. From the empty framework to the guest-loaded framework at 126 cc/g, the equilibrium volume shifts from 744 Å$^3$ to 766 Å$^3$. Furthermore, as seen from the pressure profiles in Fig. 5d, the presence of guest molecules removes the possibility of a metastable cp phase being reached under the application of mechanical pressure. This can be rationalized from the lower possible pore volume in the cp phase, hindering the presence of guest molecules. However, these conclusions could change when loading the framework with water instead as the oxalic acid linkers could interact strongly with present water adsorbates, possibly even stabilizing the lower-volume cp phase instead of destabilizing it, as is the case for carbon dioxide. Subsequently, we expanded this investigation by including $H_2O$ guests as water was suspected to be able to stabilize lower-volume states of the framework more than $CO_2$. The same simulations as $CO_2$ were performed for $H_2O$ as guest at a range of loadings. The results of these additional simulations are shown in Fig. 5e, f.

Comparing the free energy profiles of the $CO_2$-loaded and the $H_2O$-loaded framework reveals some interesting differences. First of all, low $H_2O$ loadings indeed stabilized lower framework volumes, where this is not seen for $CO_2$. This stabilization also affects the transition pressure that would be required to trigger a phase transition from the lp to cp phase. At the lowest loading of 15.7 cc/g, the transition pressure is lowered to 293 MPa, compared to 376 MPa for the empty framework.

Moreover, a larger spread of lp volumes is predicted under water adsorption compared to $CO_2$. However, for water, a non-monotonic behavior is observed. The free energy minimum first decreases from 744 Å$^3$ to 725 Å$^3$ at intermediate loadings (31.5 cc/g), and subsequently increases to 764 Å$^3$ at the highest loading. This demonstrates how, at intermediate water loadings, lower volumes are stabilized. Only when further increasing the water loadings are larger volumes again stabilized. This is consistent with our hypothesis that water, through its stronger interactions with the oxalate linkers, can more effectively stabilize the lower framework volumes than $CO_2$. Simulated PXRD patterns for guest-induced CALF-20 structures are shown in Supplementary Fig. 1. Similar to experimental PXRDs shown in Fig. 3a, simulated PXRDs also clearly show global flexibility for $CO_2$- and water-induced CALF-20 structures, demonstrated by the different patterns obtained from five representative framework snapshots during the NPT simulations.

From the free energy calculations, we also analyzed the rotation of the triazole linkers in the empty, $CO_2$-loaded and $H_2O$-loaded CALF-20 structures. Essentially, we considered the angle between the normal on the plane defined by the linker and the YZ-plane, which determines how much the linker is rotated to "obstruct" the pore along the X-axis (see Supplementary Fig. 2). For the experimental structure, this angle is ca. 21° whereas the calculation of the angle distributions for the empty, $CO_2$-loaded and $H_2O$-loaded CALF-20 structures yields a wide range of values up to 60°. Interestingly, the $H_2O$-loaded structure exhibit larger angles with peaks of around 30–40° compared to those for the empty (10–20°) and $CO_2$-loaded (20–30°) structures. This means that with water loading, the triazole linker is further rotated compared to the $CO_2$-loaded structure. This finding suggests the high $CO_2$ selectivity of the framework with respect to water, as water likes a more rotated triazole angle to optimize its interactions with the framework. However, when $CO_2$ is present, this rotation is somewhat inhibited.

## Water adsorption in CALF-20

Water is a ubiquitous component in flue gas and often adversely affects the efficiency of adsorbents because it can be preferentially adsorbed over $CO_2$. This phenomenon depends on the relative binding affinity of $CO_2$ and water with the adsorbent which is related to the heat of adsorption. The heat of adsorption for water is often high and the initial adsorbed molecules can seed and attract more water molecules through formation of strong intermolecular hydrogen bonds. This often results in a sudden and sharp increase in the water uptake reaching saturation. To evaluate the water-framework and water-water interactions at different levels of humidity, we began by simulating water adsorption in CALF-20 and compared the results with experiments at 293 K (Fig. 6a). We found a reasonably good agreement between the experiments and simulations throughout the entire pressure range. In particular, the simulations predict the shape of the type-V isotherm typical of hydrophobic adsorbents, with poor water-sorbent interactions and relatively stronger intermolecular attraction. Both simulations and experiments show the inflection point below 20% RH with saturation loading of around 11 mmol/g: these findings are similar to water adsorption data previously reported by Lin et al.[13]. Supplementary Fig. 3a shows the predicted heat of adsorption versus water loading in molecules/unit cell of CALF-20. Overall, the heat of adsorption increases as more water molecules adsorbed in CALF-20 from ca. 37–40 kJ/mol at low loadings (1 molecules/unit cell) to ~50 kJ/mol at higher loadings of ca. 8 molecules/unit cell. We also investigated the breakdown of water-water and water-framework

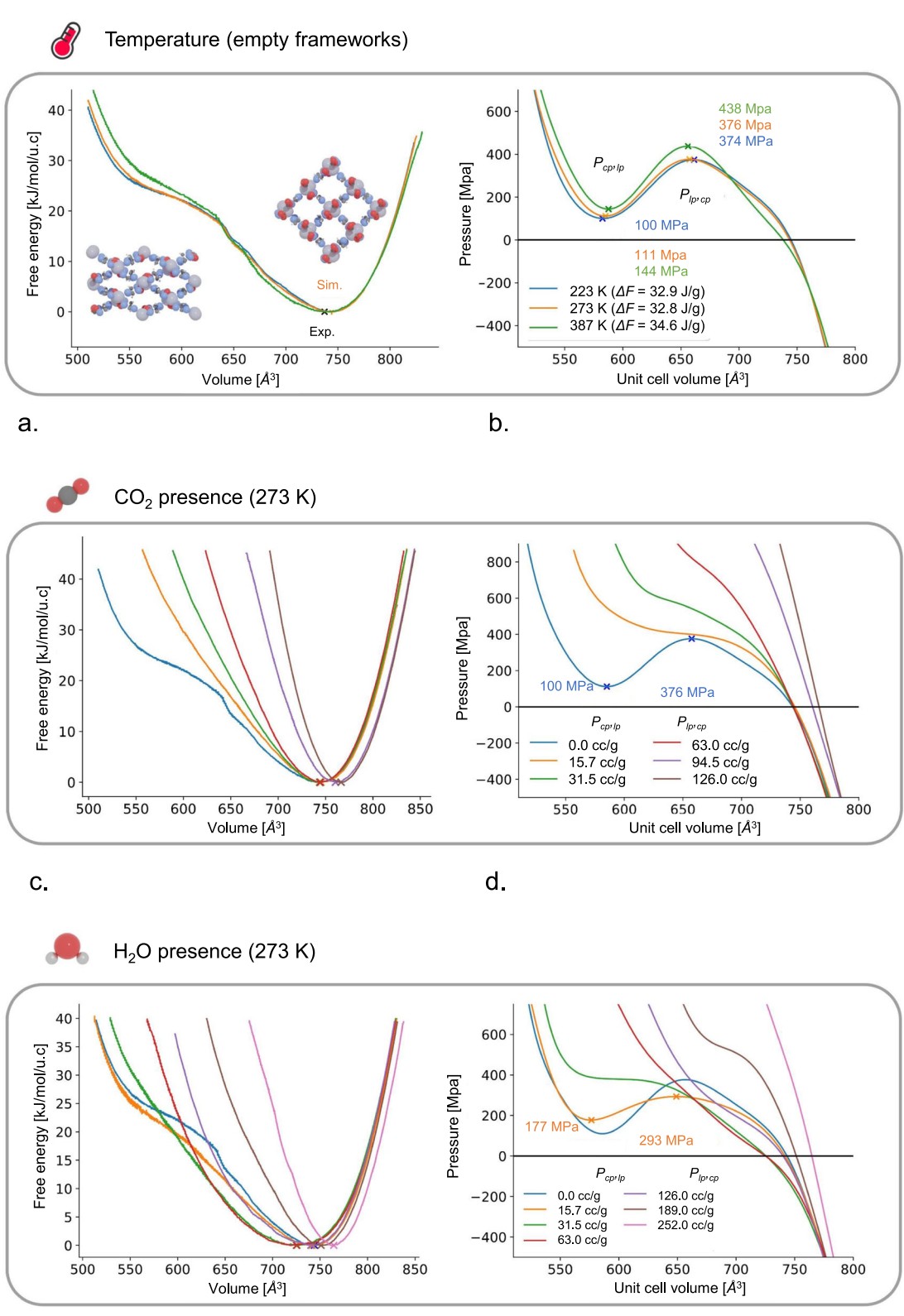

**Fig. 5 | Free energy profiles of the empty and guest-loaded CALF-20. a** Free energy profiles as a function of the unit cell volume of the empty CALF-20 framework at temperature of 223 K, 273 K, and 387 K. The inset shows closed pore (cp) and large pore (lp) structures. **b** Internal pressure of the framework, calculated as the negative derivative of the free energy, revealing the possibility of a metastable cp phase at a volume of 585 Å³; Free energy profiles of **c** $CO_2$-loaded and **e** water-loaded CALF-20 framework at different loadings; Internal pressure of the framework in **d** $CO_2$-loaded CALF-20; **f** water-loaded CALF-20.

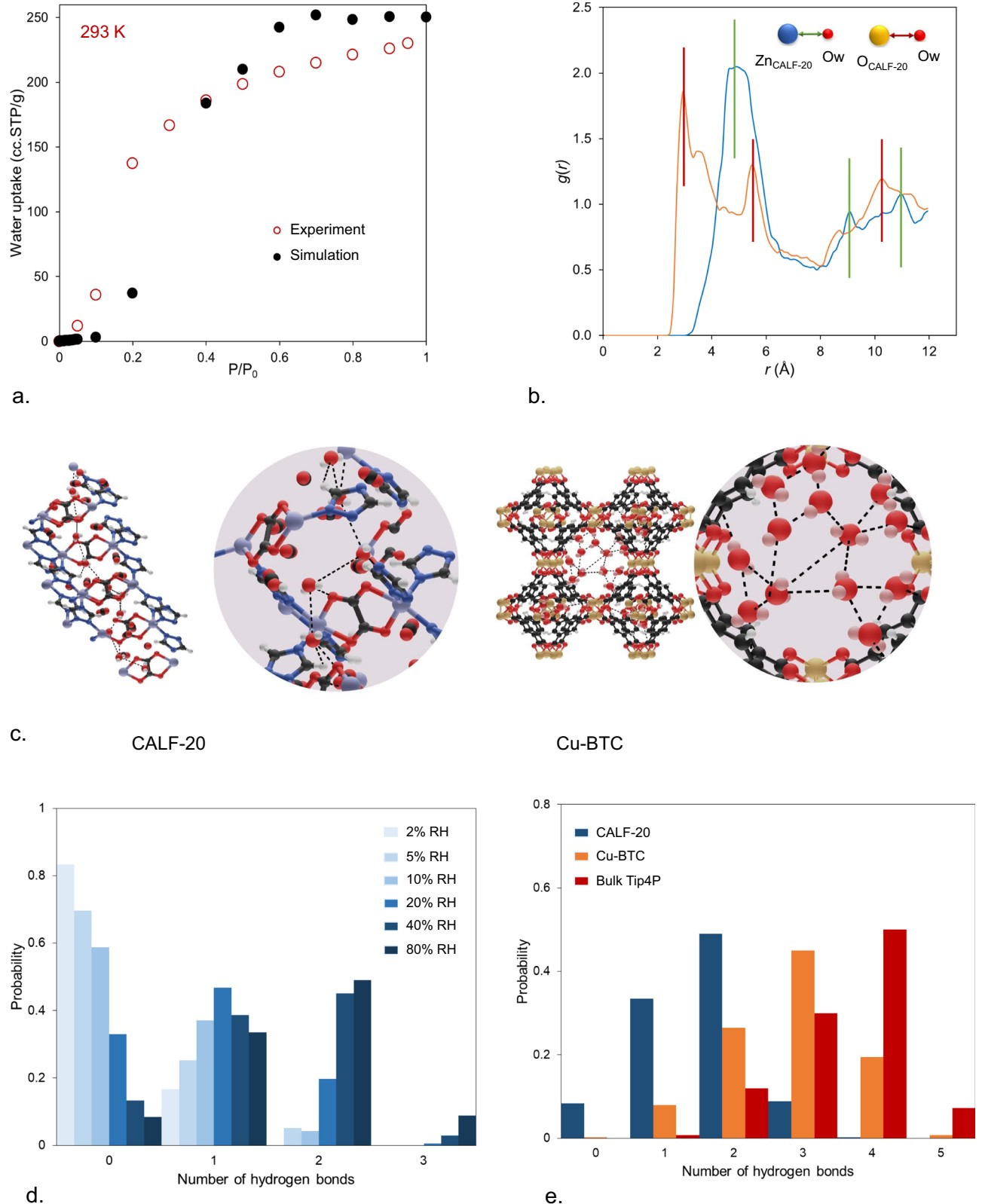

**Fig. 6 | Characterization of water adsorption and hydrogen bonds in CALF-20.**
**a** Simulated and experimental water adsorption isotherms in CALF-20 at 293 K.
**b** Radial distribution functions between framework Zn and O atom and O atom in water for 10% RH and 293 K. **c** Simulation snapshot at 10% RH (oxygen, red ball and stick representation; hydrogen, white; carbon, gray; nitrogen, blue; zinc, purple) and water molecule clustering comparison between CALF-20 and Cu-BTC.
**d** Distribution of the number of hydrogen bonds for different levels of relative humidity (RH) in CALF-20. **e** Distribution of the number of hydrogen bonds for water at 80% RH for CALF-20 compared with water in Cu-BTC and bulk TIP4P liquid water.

van der Waals (vdW) and electrostatic interactions for water adsorption in CALF-20 (Supplementary Fig. 3b, c). Electrostatic interactions account for ca. 73% of the total energy when water-MOF interactions are compared (Supplementary Fig. 3c). Electrostatic interactions between water molecules are also dominant and increase from 5 kJ/mol to around 30 kJ/mol when the RH increases from 5% to 80%.

To investigate water adsorption sites in CALF-20, we analyzed the simulation snapshots for water and studied the distance between water molecules and CALF-20's Zn and O atoms through analysis of the Radial Distribution Function (RDF). Figure 6b compares the RDF of atom pairs between Zn and O atoms in CALF-20 with the O atom in water at 10% RH where the initial water molecules are adsorbed. The first peak appears at a distance of 2.8 Å and corresponds to the distance between O of water and O of CALF-20 demonstrating that water molecules sit next to the oxygen atoms from the oxalate linkers. The distance between the Zn, and O of water occurs at larger distances of ca. 4 Å which indicates lack of direct contact with the metal atoms and explains why the material does not adsorb significant amount of water at low levels of humidity. Figure 6c shows the simulation snapshot for water adsorption in CALF-20 at 10% RH. Water molecules are adsorbed close to the oxalate pillars of CALF-20 rather than the metal clusters in agreement with the RDF results. To better understand the water adsorption mechanism in CALF-20 we studied how water forms clusters in the pores of CALF-20 and compared it with water adsorption behavior in Cu-BTC, a representative hydrophilic MOF. To achieve this, we calculated the distribution of hydrogen bonds at different relative humidities averaged over the production cycles. To calculate the number of hydrogen bonds, we used a geometric criterion described by Xu et al.[37]. In these calculations, a pair of water molecules is considered hydrogen bonded if the O-O distance is below 3.5 Å and simultaneously the O-H...O angle is greater than 150°. Using this criterion we obtained the hydrogen bond distributions for different water loadings in CALF-20 shown in Fig. 6d. At relative humidities less than 20%, water molecules are far apart and form zero or, at most, one hydrogen bond. At higher relative humidities (80%), after condensation occurs, water molecules start to interact but only begin to form one or two hydrogen bonds per water molecule. This indicates that water molecules are more spread out in the pore space and clustering becomes less probable within the small pores of CALF-20. However, at high RH values in Cu-BTC, we observe up to three and four hydrogen bonds per water molecule, indicating a strong tendency for clustering (Fig. 6e). In liquid water (red), molecules construct mainly four hydrogen bonds, forming tetrahedral conformations of water clusters. In general, when compared with bulk water, we observe a reduction in the dominant number of hydrogen bonds in the adsorbed phase from four to three in Cu-BTC, and from four to two in CALF-20. These findings are also supported by the water adsorption snapshots displayed in Fig. 6c where the hydrogen bonds are schematically illustrated. Overall, the hydrophobicity observed in CALF-20 at low RH is related to the lack of sufficient force from water to open the framework and accommodate enough water to favor direct cooperative contact between water molecules, the lack of water's direct contact with the Zn atoms, as well as pore confinement effects for which strong hydrogen bonding between neighboring water molecules cannot occur.

In summary, CALF-20 is an outstanding water stable structure capable of selectively separating $CO_2$ from flue gas. Here, we provide an in-depth study of the gas adsorption properties and framework flexibility of CALF-20 combining different simulation and experimental techniques. The unexpected underprediction of $CO_2$ adsorption, when compared with experiments, suggested structural changes in the presence of gas molecules most notably at 273 K. CALF-20's framework flexibility was explored using experimental gas adsorption and PXRD data in combination with molecular dynamics simulations at the DFT level. At 273 K, the difference between the pore volume fractions of the empty and $CO_2$-loaded framework was calculated to be ca. 12% demonstrating that the adsorbed guests clearly increase the pore

space for more adsorbates, consistent with the hypothesis that the difference between the experimental observations and the Monte Carlo simulations is due to the flexibility of the framework under guest-adsorption. Furthermore, the complete free energy profiles of the empty and guest-loaded frameworks were computed, making use of machine learning potential (MLPs) trained to enhanced sampling DFT data, demonstrating the induced flexibility of the framework under guest adsorption. We note that, our approach of generating training data for an MLP at the DFT level to fully characterize the framework flexibility as a function of temperature and guest adsorption has the promise to be extended widely to other nanoporous materials. The investigation of guest-induced framework flexibility at the DFT level has mostly been limited to energy optimizations[38]. In contrast, trained MLPs can be used at a significantly reduced computational costs, allowing for the first principles construction of temperature and guest-loading dependent free energy profiles. In this way, the behavior of MOFs can be characterized at the relevant operating conditions. Furthermore, recent implementations of active learning loops for MLP training can further reduce the required number of DFT evaluations, enabling wide-scale applications to the field of nanoporous materials[39]. The investigation into the hydrophobic nature of CALF-20 showed that water molecules do not interact directly with the Zn and instead prefer to sit inside the small pores, as evidenced by simulation snapshots and radial distribution function analysis. The analysis of the hydrogen bond network showed that water molecules are spread out in the tightly confined pores of CALF-20 which inhibits formation of more than two hydrogen bonds per water molecule and therefore water clustering is less probable. In conclusion, we demonstrated a great example of collaboration and feedback between computational and experimental MOF researchers to encourage identification and characterization of other hydrophobic MOF materials useful for $CO_2$ capture applications.

## Methods

### GCMC simulations of gas adsorption in CALF-20
Gas adsorption simulations were carried out via the grand canonical Monte Carlo (GCMC) calculations performed in RASPA-2.0 code[40]. In the GCMC simulations, insertion, deletion, and translation and rotation moves were attempted with equal probability. The interactions between the gas species and the framework were modeled using Lennard–Jones (LJ) plus Coulomb potentials. LJ parameters for all atoms in MOFs were taken from Dreiding force field (DFF)[41] and were truncated at a cutoff radius of 12.8 Å. The force field parameters for the adsorbates and CALF-20 are tabulated in Supplementary Tables 1–4 in the supporting information. The Lorentz-Berthelot mixing rules were used to calculate cross interactions. Partial atomic charges for CALF-20 were calculated using the REPEAT (Repeating Electrostatic Potential Extracted Atomic) method[42] and the Ewald summation technique was used to calculate electrostatic interactions. GCMC simulations for $N_2$ and $CO_2$ adsorption were run for 20,000 cycles for equilibration and a further 20,000 cycles to average properties. $N_2$ and $CO_2$ were modeled using the TraPPE model[43]. For water simulations we used 100,000 cycles for equilibration and subsequent 100,000 cycles for production. Water was modeled using the TIP4P force field[44]. The relative humidity of 100% corresponds to the saturation pressure of the TIP4P model.

### (MLP) MD simulations of the empty and guest-loaded CALF-20
To assess the flexibility of the CALF-20 framework, NPT MD simulations with a fully flexible unit cell were performed for both the empty and guest-loaded frameworks at 273 K, 323 K, and 387 K with the PBE-D3(BJ) level of theory using the CP2K software package (version 7.1)[33]. A plane wave energy cutoff of 500 Ry and GTH pseudopotentials were used, employing the TZVP-MOLOPT basis set. Simulations were performed for a duration of 20 ps.

To compute the free energy profiles of the empty and guest-loaded framework, machine learning potentials (MLPs) were trained and employed to significantly reduce the required computational resources. First, DFT metadynamics simulations are performed for 2 ps on the $2 \times 2 \times 2$ CALF-20 supercell (both empty framework and guest-loaded frameworks) with the cell volume as a collective variable to explore the space of possible states of the framework. In these simulations, Gaussian hills with a height of 10 kJ/mol and a width of 50 Å$^3$ are added every 25 fs. For both the empty and guest-loaded framework, short (0.5 ps) DFT umbrella sampling (US) simulations of the $2 \times 2 \times 2$ supercells were then performed, restrained with a bias potential with a strength of 0.005 kJ/Å$^6$ at volumes between 4000 Å$^3$ and 7000 Å$^3$ with a step of 50 Å$^3$. PLUMED was used to apply the bias potential[45,46].

The energy and forces of snapshots extracted from these simulations every 5 fs were then used to train a NequIP model[36]. A model was trained separately for each guest-loading. The dataset was randomly divided into a training and validation set with a 90:10 ratio. The MLPs were trained making use of a cutoff radius of 5 Å, 4 interaction blocks, a maximum rotation order of 1 and 64 features. The loss function contains both energies (with weight 1) and forces (with weight 5). For all MLPs, a validation error lower than 0.13 meV per atom for energies and 39.6 meV/Å for forces were obtained. With the trained MLPs, US MLP MD simulations were performed, restrained at the same set of volumes as performed with CP2K[33], for a duration of 15 ps. To obtain the unbiased free energy profiles, the weighted histogram analysis method (WHAM) was used[47]. The internal pressure as a function of the volume was derived by fitting a 12th order polynomial to the free energy profiles, and computing the negative derivative with respect to the volume[35].

## Data availability

Details of the GCMC, MD, DFT and MLP calculations along with experimental methods are outlined in the Methods section and the Supplementary Information. All data points supporting the findings of this study are freely accessible within the paper and its Supplementary Information including PXRD patterns for guest-loaded CALF-20 structures (XLSX); Water adsorption energies in CALF-20 and force field parameters (PDF). Source data are provided with this paper. Original images of CALF-20 structures, its topology and adsorption snapshots are also available on request, which can be addressed to Peyman Z. Moghadam. Source data are provided with this paper.

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

## Acknowledgements

P.Z.M. acknowledges support from the Department of Science, Innovation and Technology (DSIT) and the Royal Academy of Engineering under the Industrial Fellowships programme (IF2223–110). He also acknowledges support from the Engineering and Physical Science Research Council (EPSRC) and the Cambridge Crystallographic Data Center (CCDC) for the provision of Ph.D studentship funding to L.T.G. The authors also acknowledge financial support from the Research Board of Ghent University (BOF). The computational resources (Stevin Supercomputer Infrastructure) and services provided by the VSC (Flemish Supercomputer Center), funded by Ghent University, FWO, and the Flemish Government, department EWI. R.O. acknowledges funding support during his Ph.D study from the Indonesian Endowment Fund for Education (LPDP) with contract no. 202002220216006.

## Author contributions

‡ R. O. and R. G. contributed equally to the paper. R. O., R. G., and L. T. G. performed the simulations. P. S., R. H., O. T. G, O. G. N, and N. M. planned and carried out experimental studies including PXRD and gas adsorption measurements. P. H., J. L. C., V. V. S and P. Z. M conceptualized and supervised the project. We also thank Ramanathan Vaidhyanathan for reviewing the paper. All authors reviewed and edited the manuscript.

## Competing interests

Svante Inc. has licensed two CALF-20 patents (CA2904546A1 and EP3784824A1) for $CO_2$ capture application. The remaining authors declare no competing interests.
