## [Peer Review File · Nature Communications]

Gas adsorption and framework flexibility of CALF-20 explored via experiments and simulationsReviewers' Comments:

Reviewer #1:

Remarks to the Author:

The manuscript, by Moghadam and co-workers, reports the computational and experimental study of the framework flexibility of a commercialized MOF — CALF-20. Although the authors implemented state-of-the-art techniques in the computational part, this is a technical heavy paper in its current form and required a major revision to meet the high standard and scholarly presentation of Nature Communications.

Several suggestions are provided as follows:

1. While the computational techniques employed are commendable, additional rigorous experimental work is crucial to bolster your conclusions. The manuscript leans heavily towards commercialization in the introduction, making it challenging for readers to discern between established knowledge and new insights. Comparisons with Shimizu's Science paper on water adsorption mechanisms should be included to clarify the novelty of your work.
2. In-situ PXRD patterns, especially with varying CO₂ concentrations and humidity, are essential for a detailed understanding of structural transformations. Geometrical variances of the frameworks, including metal centres, chelating linkers, and the metal-triazolate layer, should be defined to elucidate how frameworks change during water and CO₂ adsorption.
3. Reformat the manuscript and redraw figures according to Nature's standards. In Figure 1a, the C-H bond in triazolate appears too long; rectify this to meet geometrically reasonable standards. Oxalic acid should appear to be oxalate for consistency.

Minor Revisions:

1. The article mentions molecular dynamics simulations based on density functional theory (DFT) and machine learning potentials (MLPs), but fails to elaborate on the potential insights and significance of these simulation results for the design and development of MOF materials. Further discussion is needed to explore the potential impact of these simulation results on the development of other MOF materials for CO₂ capture applications.
2. The article inadequately discusses the potential limitations and uncertainties of the research findings. Therefore, it is suggested that the authors provide a more objective assessment of the research outcomes in the conclusion, including an honest discussion of potential limitations, to enhance the scientific credibility of the study.
3. The overlay of the structure diagram and simplified diagram in Figure 1b is unnecessary since the structure diagram and topological diagram are clear enough.
4. The sentence "In fact, for many MOFs, adsorption simulations tend to overestimate the amount of gas adsorbed since they are carried out on pristine crystalline materials where residual solvent is removed before calculation" needs a reference to support it. It is recommended to cite a relevant source. Additionally, it would be helpful to explain why the removal of residual solvent leads to higher simulation values than experimental values.
5. In Figure 3a, the sentence "We obtained PXRD data under different gas and liquid exposures (N₂, CO₂, water, and ethanol) and compared the results with the as-synthesized CALF-20 sample" does not match the actual content of the figure, which shows a simulated one. This description needs to be clarified.
6. In Figure 6c, it would be helpful to use different colors to distinguish atoms, as it is difficult to determine whether water molecules are interacting with the metal or acid pillars without color differentiation.

Reviewer #2:

Remarks to the Author:

The manuscript authored by Oktavian et al. provides a comprehensive exploration of the gas

adsorption properties of CALF-20, a MOF currently undergoing scale-up for post-combustion separation. The authors initially conducted a comparative analysis of experimentally and computationally derived gas adsorption isotherms, leading them to delve into host-guest interactions using N₂, CO₂, and H₂O as guest molecules. Their findings reveal structural changes induced by specific host-guest interactions, particularly noting potential H-bonding interactions when water occupies the pores of CALF-20. This work effectively demonstrates the synergy between experimental and computational approaches for a better understanding of the sorption behavior in porous MOFs. While acknowledging the credibility of the study, I have reservations concerning the motivation and conclusions. At this stage, the manuscript falls short of meeting the standards for publication in Nature Communications and may be reconsidered following major revisions. A few of my comments are listed below:

- The manuscript claims novelty in exploring the impact of H₂O on CALF-20 structure, asserting it as a first-time investigation. However, similar studies have been conducted, and the authors should revise their statements accordingly. Proper credit should be given to previous research:

ACS Appl. Nano Mater. 2023, 6, 21, 19963–19971

ACS Appl. Mater. Interfaces 2023, 15, 41, 48287–48295

ACS Materials Lett. 2023, 5, 11, 2942–2947

- The abstract and conclusions suggest that the work encourages the development of other MOFs for humid CO₂ separations, but the manuscript lacks clarity on how this encouragement is derived. The authors are encouraged to elaborate on this aspect.
- The methodology regarding PXRD patterns for CALF-20 loaded samples is unclear. The claim of experiments conducted under flow raises questions, especially as only one PXRD pattern is presented. The authors are encouraged to provide additional patterns and ensure that equilibrium is reached during experimentation. Comparison with prior research is also recommended.
- The authors are encouraged to carefully review the manuscript, as there are a few typographical errors that need correction.

Title of the paper: Gas adsorption and framework flexibility of CALF-20 explored via experiments and simulations

We thank both reviewers for the thoughtful and helpful suggestions. The reviewers' comments and the editorial requests are repeated below in *italics* along with our responses in black normal font and the corresponding changes made to the manuscript in red.

Reviewers' comments:

Reviewer #1 (Remarks to the Author):

“The manuscript, by Moghadam and co-workers, reports the computational and experimental study of the framework flexibility of a commercialized MOF — CALF-20. Although the authors implemented state-of-the-art techniques in the computational part, this is a technical heavy paper in its current form and required a major revision to meet the high standard and scholarly presentation of Nature Communications. Several suggestions are provided as follows:”

“1. While the computational techniques employed are commendable, additional rigorous experimental work is crucial to bolster your conclusions. The manuscript leans heavily towards commercialization in the introduction, making it challenging for readers to discern between established knowledge and new insights. Comparisons with Shimizu's Science paper on water adsorption mechanisms should be included to clarify the novelty of your work.”

Response: We have now conducted additional *in-situ* PXRD measurements— under various adsorption conditions— to further examine structural transformations in CALF-20. This point is fully addressed in the reviewer's next comment. We have also expanded the Introduction section and added a paragraph on page 2 to add three new references, and highlight the novelty of our work. We note that while our paper was under review by *Nat Commun*, these three papers (with a focus on CALF-20) were published in late 2023:

ACS Appl. Nano Mater. 2023, 6, 21, 19963–19971
ACS Appl. Mater. Interfaces 2023, 15, 41, 48287–48295
ACS Materials Lett. 2023, 5, 11, 2942–2947

Compared to the above studies and Shimizu's Science paper, the analysis of CALF-20's structural transformations via *in-situ* adsorption/PXRD data (see new data from Svante) in combination with first-principles molecular dynamics (MD) simulations, derivation of machine learning potentials (MLPs) trained to DFT data, and the analysis of distribution of water hydrogen bonds at different relative humidities, are extremely novel. We note that generation of training data for MLPs at the DFT level to characterize the framework flexibility in response to external stimuli (e.g. temperature or adsorption) has great promise to be extended widely to other nanoporous materials and their characterization for adsorption applications. We also note that all the computations presented in our work were performed before laboratory synthesis and gas

adsorption measurements were carried out by Svante, and the good agreement between simulations and experiments provided a powerful example of the predictive ability of molecular modelling for the assessment of MOFs for CO₂ capture in humid conditions.

Please see the revised discussion on Page 2: In 2021, Svante reported single-step commercial synthesis of CALF-20 for Temperature Swing Adsorption (TSA) processing of up to 1 tonne of CO₂ removal per day from cement flue gas.²¹ CALF-20 has now become the hallmark of success among MOFs undergoing the scale up process from laboratory to industry - as it ticks many of the required boxes for an optimum adsorbent for practical CO₂ capture. Developing new materials for CO₂ abatement has never been more critical, and in this context, a number of research groups have started examining CALF-20 in more detail with the aim of aiding the design and development of other adsorbent materials for selective adsorption of CO₂. In late 2023, Ho and Paesani²² and Magnin et al.²³ studied competitive adsorption and diffusion of water and CO₂ in CALF-20 via classical Monte Carlo and molecular dynamics simulations. At the same time, Chen et al.²⁴ looked into structural transformations of CALF-20 in humid environments via powder X-ray diffraction (PXRD) and pair distribution function analysis. In the present study, in collaboration with MOF scientists from Svante, we used a close feedback loop between simulations and experiments to obtain molecular-level insights into some of the key CO₂ adsorption properties of CALF-20, and determine its structural flexibility triggered by the presence of guest molecules. We investigated water-CALF-20 interactions through water adsorption simulations and hydrogen-bond analysis, and studied the structural transformations of CALF-20 using *in-situ* adsorption/PXRD data in combination with first-principles molecular dynamics (MD) simulations. With the aid of machine learning potentials (MLPs) trained to DFT data, fully converged free energy profiles of both the empty and guest-loaded frameworks were also computed, demonstrating the guest-induced flexibility of the framework. Importantly, in this work, we note that all the computations were performed before laboratory synthesis and physical gas adsorption measurements were carried out by Svante. The excellent agreement between simulation and experiment provided a powerful example of the predictive ability of molecular modelling, showcased in the assessment of MOF candidates for CO₂ capture in wet conditions.

“2. In-situ PXRD patterns, especially with varying CO₂ concentrations and humidity, are essential for a detailed understanding of structural transformations. Geometrical variances of the frameworks, including metal centres, chelating linkers, and the metal-triazolate layer, should be defined to elucidate how frameworks change during water and CO₂ adsorption.”

Response: To address the reviewer’s comment, in the past two months, we have conducted additional *in-situ* PXRD experiments to further examine CALF-20’s structural transformations. We have obtained PXRD data under different relative humidities of 4%, 10% and 20%, as well as exposure to liquid water, ethanol and other gases such as N₂, and CO₂. All new PXRD patterns are shown in Figure 3a and compared with the simulated CALF-20 sample.

Pages 3-4: The CIF from the published CALF-20 structure contains EtOH as guest molecules. Hence, when the experimentally synthesized CALF-20 powder is soaked in EtOH and then briefly dried (see SI for information), the PXRD obtained matches extremely well with the simulated one. Since solvents induced substantial changes to the PXRD pattern, we evacuated

CALF-20 by drying the powder in-situ at 110°C for 45 minutes under N₂ flow (Figure 3a), see SI for information.

The evacuated PXRD pattern is similar to the simulated or EtOH soaked sample, with only slight changes in the (100) and (011) reflections. The (100) plane in the simulated pattern, which corresponds to the interplanar distances between the Zn-triazolate layers in the yz-plane (Figure 3b), exhibits a slight shift to lower 2θ and expansion of this plane when the sample is evacuated. There is a corresponding shift in the (011) reflection, which represents the middle of the pore (Figure 3c), to a higher 2θ of the evacuated sample, suggesting a contraction along this plane. This implies that in the absence of solvent, the pores of CALF-20 as viewed along the x axis are contracted, with a corresponding expansion between the Zn-triazolate layers compared to the EtOH soaked CALF-20. We take this evacuated CALF-20 as a baseline for further guest loaded studies.

Given CALF-20's ability to selectively adsorb CO₂ over water at low relative humidity (RH), we loaded CALF-20 with water and CO₂ to observe the effects on the PXRD pattern (see SI for experimental details). Compared to the evacuated sample, loading CALF-20 with 100% CO₂ and soaking in liquid water induced a slight shift in the (100) reflection to higher 2θ, showing contraction between the Zn-triazolate layers, similar to what is observed in the EtOH soaked sample. In addition, the (020) reflection corresponding to the planes containing the oxalate moieties (Figure 3d) shifts to lower 2θ for both water and CO₂, and it does not show much change for ethanol. In contrast, the (011) reflection remains the same in the CO₂ loaded sample, whereas a shift to lower 2θ is observed in the water soaked sample. The PXRD pattern of CALF-20 does not show significant changes at low RH (4% and 10%). At 20% RH, an obvious change of pattern was observed at (100), (011) and (020) planes. This is consistent with the reported phase change of CALF-20 in moisture.²⁴ Thus, H₂O accommodation re-quires the (011), (100) and (020) peaks to shift significantly, while CO₂ requires only the (100) and (020) reflections to move. The differences between PXRD patterns at 20%RH and water soaking indicated that water soaking might bring different changes to the structure compared to the adsorption of water vapor. Overall, water is able to bring more changes to the framework compared to all other guests discussed here. In other words, CO₂'s accommodation requires less structural change than H₂O; this could explain why CALF-20 accepts CO₂ over water at low RH. A simplified explanation would be that CO₂ adsorption into CALF-20 does not require much structural adjustment of the activated phase, whereas water needs the framework to open slightly more: even before there is sufficient water to make this structural change, the CO₂ fills up, making CALF-20 more CO₂ selective. These changes in the PXRD patterns demonstrate that CALF-20 is not a rigid and undergoes structural flexibility dependent on the guests inside the pores.

Figure 3. **a.** Comparison of PXRD patterns for CALF-20 at different adsorption conditions, and for fully dried samples under dry N₂ and CO₂ at 50 °C. For the samples under gas flow, CALF-20 was first heated to 110 °C for 30 minutes under dry N₂ flow to evacuate the sample. Then, the sample was cooled to 50 °C and the PXRD pattern was collected under N₂ or CO₂ flow at 50 °C. PXRD patterns under different relative humidities (%RH) were all collected in N₂. CALF-20 structure with highlighted **b.** (100), **c.** (011) and **d.** (020) *hkl* planes.

We note that Figure 4 presents the distribution of geometric properties (pore volume and cell length) of empty and guest-loaded CALF-20 from the MD simulations at different temperatures. In the revised manuscript, we further analysed the rotation of the triazolite linkers in the empty, CO₂-loaded and H₂O-loaded CALF-20 structures. We have now added this discussion on page 7 and Figure S2 in the supporting information.

Page 7: From the free energy calculations, we also analysed the rotation of the triazolite linkers in the empty, CO₂-loaded and H₂O-loaded CALF-20 structures. Essentially, we considered the angle between the normal on the plane defined by the linker and the YZ-plane, which determines how much the linker is rotated to "obstruct" the pore along the X-axis (see Figure S2). For the experimental structure, this angle is ca. 21° whereas the calculation of the angle distributions for the empty, CO₂-loaded and H₂O-loaded CALF-20 structures yields a wide range of values up to 60°. Interestingly, the H₂O-loaded structure exhibit larger angles with peaks of around 30-40° compared to those for the empty (10-20°) and CO₂-loaded (20-30°) structures. This means that with water loading, the imidazolate linker is further rotated compared to the CO₂-loaded structure. This finding suggests the high CO₂ selectivity of the framework with respect to water, as water likes a more rotated imidazole angle to optimize its interactions with the framework. However, when CO₂ is present, this rotation is somewhat inhibited.

Figure S2. a. Histograms showing the rotation of the triazole linkers in the experimental, empty, CO₂-loaded and H₂O-loaded CALF-20 structures. The angle distributions are obtained from the free energy profiles at saturation loadings and 273 K. b. Schematic showing the angle between the normal on the plane defined by the triazole linker and the YZ-plane calculated in a

“3. Reformat the manuscript and redraw figures according to Nature's standards. In Figure 1a, the C-H bond in triazolate appears too long; rectify this to meet geometrically reasonable standards. Oxalic acid should appear to be oxalate for consistency.”

Response: We thank the reviewer for pointing this out. We have now redrawn the Figures.

Minor Revisions

“1. The article mentions molecular dynamics simulations based on density functional theory (DFT) and machine learning potentials (MLPs), but fails to elaborate on the potential insights and significance of these simulation results for the design and development of MOF materials. Further discussion is needed to explore the potential impact of these simulation results on the development of other MOF materials for CO₂ capture applications.”

Response: The detailed discussion for the simulation results are now added to the manuscript to explore the potential of further development of MOF materials for CO₂ capture application. We also agree with the reviewer that the computational techniques employed in this work have relevance not only to this work, but to future investigations on the interplay between adsorption and framework flexibility. Therefore, we have added the following discussion in the conclusions:

“We note that, our novel approach of generating training data for an MLP at the DFT level to fully characterize the framework flexibility as a function of temperature and guest adsorption has the promise to be extended widely to other nanoporous materials. The investigation of guest-induced framework flexibility at the DFT level has mostly been limited to energy optimizations.⁴⁴ In contrast, trained MLPs can be used at a significantly reduced computational costs, allowing for the first principles construction of temperature and guest-loading dependent free energy profiles. In this way, the behavior of MOFs can be characterized at the relevant

operating conditions. Furthermore, recent implementations of active learning loops for MLP training can further reduce the required number of DFT evaluations, enabling wide-scale applications to the field of nanoporous materials.”

“2. The article inadequately discusses the potential limitations and uncertainties of the research findings. Therefore, it is suggested that the authors provide a more objective assessment of the research outcomes in the conclusion, including an honest discussion of potential limitations, to enhance the scientific credibility of the study.”

Response: Please see our response to the comment above. To further address the reviewer’s comment, we have also revised the manuscript with additional information on page 4:

Simulation of MOFs exhibiting structural flexibility is challenging. Our reported GCMC simulations of gas adsorption (Figure 2) made the assumption that framework atoms are fixed at their crystallographic positions, thus we modelled CALF-20 as rigid. This assumption is valid for many MOFs whose building block topology do not allow for high degrees of flexibility. However, simulated gas adsorption predictions can deviate from experiments when MOFs are structurally flexible in response to external stimuli such as temperature or guest loading.³⁰ Here, to further investigate the effects of adsorption-induced flexibility in CALF-20, we also performed MD simulations at the density-functional (DFT) level of theory (PBE-D3(BJ)) using CP2K. By comparing MD simulations of both the empty and the guest-loaded CALF-20 framework at experimentally observed loadings, the effect of guests on the framework can be directly determined.

“3. The overlay of the structure diagram and simplified diagram in Figure 1b is unnecessary since the structure diagram and topological diagram are clear enough.”

Response: The overlay of the structure diagram is now removed.

“4. The sentence "In fact, for many MOFs, adsorption simulations tend to overestimate the amount of gas adsorbed since they are carried out on pristine crystalline materials where residual solvent is removed before calculation" needs a reference to support it. It is recommended to cite a relevant source. Additionally, it would be helpful to explain why the removal of residual solvent leads to higher simulation values than experimental values.”

Response: We have now revised this section on page 3 to make it more clear and add the relevant references.

Generally, the experimentally synthesized MOFs contain solvents in their pores, which can be removed upon activation. Before performing gas adsorption simulations, these solvent molecules can be fully removed mimicking the experimental activation process. This process assumes that the experimental activation is successful in removing all residual solvent inside the pores and the structure is not changed upon removing the solvent. Clearly, incomplete experimental activation in MOFs can reduce the accessibility to the pore space. Therefore, when solvent-free structures are used in simulations, the amount of predicted gas adsorption is usually higher than experimental measurements, given that more pore space is accessible for guest molecules.^{29,30}

Since the atoms in CALF-20's structure are fixed at their crystallographic positions during the simulations, underpredicting CO₂ uptake indicated to us that CALF-20 undergoes structural changes in the presence of external stimuli such as gas adsorption or temperature.

"5. In Figure 3a, the sentence "We obtained PXRD data under different gas and liquid exposures (N₂, CO₂, water, and ethanol) and compared the results with the as-synthesized CALF-20 sample" does not match the actual content of the figure, which shows a simulated one. This description needs to be clarified."

Response: We thank the reviewer for pointing this out. We have now changed the text to clarify this point.

"6. In Figure 6c, it would be helpful to use different colors to distinguish atoms, as it is difficult to determine whether water molecules are interacting with the metal or acid pillars without color differentiation."

Response: We have now used different colors to represent the CALF-20 framework in Figure 6c.

C.

CALF-20

Cu-BTC

Reviewer #2 (Remarks to the Author):

"The manuscript authored by Oktavian et al. provides a comprehensive exploration of the gas adsorption properties of CALF-20, a MOF currently undergoing scale-up for post-combustion separation. The authors initially conducted a comparative analysis of experimentally and computationally derived gas adsorption isotherms, leading them to delve into host-guest interactions using N₂, CO₂, and H₂O as guest molecules. Their findings reveal structural changes induced by specific host-guest interactions, particularly noting potential H-bonding interactions when water occupies the pores of CALF-20. This work effectively demonstrates the synergy between experimental and computational approaches for a better understanding of the sorption behavior in porous MOFs. While acknowledging the credibility of the study, I have reservations concerning the motivation and conclusions. At this stage, the manuscript falls short

of meeting the standards for publication in Nature Communications and may be reconsidered following major revisions. A few of my comments are listed below:”

*“The manuscript claims novelty in exploring the impact of H₂O on CALF-20 structure, asserting it as a first-time investigation. However, similar studies have been conducted, and the authors should revise their statements accordingly. Proper credit should be given to previous research: ACS Appl. Nano Mater. 2023, 6, 21, 19963–19971
ACS Appl. Mater. Interfaces 2023, 15, 41, 48287–48295
ACS Materials Lett. 2023, 5, 11, 2942–2947”*

Response: We thank the reviewer for their suggestion. Given the industrial importance of CALF-20 for CO₂ capture, there are indeed a number of groups across the world who have started working on this structure. We note that all three papers suggested by the reviewer, were published in late 2023 when our paper was under review by *Nat Commun*. In response to this comment and that of reviewer 1 (comment #1), we have now expanded the Introduction section and added the paragraph on page 2 to add these references, and highlight the novelty of our paper. We avoid repeating our response here.

“The abstract and conclusions suggest that the work encourages the development of other MOFs for humid CO₂ separations, but the manuscript lacks clarity on how this encouragement is derived. The authors are encouraged to elaborate on this aspect.”

Response: We have now addressed this comment throughout the revised manuscript. In particular, the last section of the paper explicitly states this important point:

Page 10: “We note that, our novel approach of generating training data for an MLP at the DFT level to fully characterize the framework flexibility as a function of temperature and guest adsorption has the promise to be extended widely to other nanoporous materials. The investigation of guest-induced framework flexibility at the DFT level has mostly been limited to energy optimizations.⁴⁶ In contrast, trained MLPs can be used at a significantly reduced computational costs, allowing for the first principles construction of temperature and guest-loading dependent free energy profiles. In this way, the behavior of MOFs can be characterized at the relevant operating conditions. Furthermore, recent implementations of active learning loops for MLP training can further reduce the required number of DFT evaluations, enabling wide-scale applications to the field of nanoporous materials.⁴⁷ The investigation into the hydrophobic nature of CALF-20 showed that water molecules do not interact directly with the Zn and instead prefer to sit inside the small pores, as evidenced by simulation snapshots and radial distribution function analysis. The analysis of the hydrogen bond network showed that water molecules are spread out in the tightly confined pores of CALF-20 which inhibits formation of more than two hydrogen bonds per water molecule and therefore water clustering is less probable. In conclusion, we demonstrated a great example of collaboration and feedback between computational and experimental MOF researchers to encourage identification and characterization of other hydrophobic MOF materials useful for CO₂ capture applications.”

“The methodology regarding PXRD patterns for CALF-20 loaded samples is unclear. The claim of experiments conducted under flow raises questions, especially as only one PXRD pattern is presented. The authors are encouraged to provide additional patterns and ensure that equilibrium is reached during experimentation. Comparison with prior research is also recommended.”

Response: Our colleagues at Svante have conducted additional *in-situ* PXRD measurements under various adsorption conditions. Details can also be found in the response to Reviewer 1. Experimental procedures are also specified in the supporting information.

We also note that all the PXRD exposure measurements were carefully conducted to ensure equilibrium is reached. Figure below shows the PXRD patterns for CALF-20 and the kinetic study we performed at 20%RH. The phase transition completed in 40 min and did not show further changes up to 280 min.

“The authors are encouraged to carefully review the manuscript, as there are a few typographical errors that need correction.”

Response: We have now thoroughly checked our manuscript and corrected the typographical errors.

Reviewers' Comments:

Reviewer #1:

Remarks to the Author:

The authors have addressed my previous concerns. Thus, I am delighted to recommend it for publication in Nature Communications at it is.

Reviewer #2:

Remarks to the Author:

The authors have put much effort into addressing the Reviewers' comments and majorly revised their manuscript with new experiments and discussion. I, therefore, propose that this manuscript can be accepted for publication in Nature Communications.